# Novel Therapeutics in Ovarian Cancer: Expanding the Toolbox

Sara Moufarrij [1] and Roisin E. O'Cearbhaill [2,3,*]

1. Gynecology Service, Department of Surgery, Memorial Sloan Kettering Cancer Center, New York, NY 10065, USA; moufarrs@mskcc.org
2. Gynecologic Medical Oncology Service, Department of Medicine, Memorial Sloan Kettering Cancer Center, New York, NY 10065, USA
3. Department of Medicine, Weill Cornell Medical College, New York, NY 10065, USA
* Correspondence: ocearbhr@mskcc.org; Tel.: +1-646-888-4227

**Abstract:** Despite high response rates to initial therapy, most patients with ovarian cancer will ultimately recur and go on to develop resistance to standard treatments. Novel therapies have been developed to overcome drug resistance and alter the tumor immune microenvironment by targeting oncogenic pathways, activating the innate immune response, and enhancing drug delivery. In this review, we discuss the current and future roles of chemotherapy, targeted agents such as poly (ADP-ribose) polymerase (PARP) inhibitors, bevacizumab, and mirvetuximab in the treatment of ovarian cancer. We explore the emerging role of therapeutic targets, including DNA repair pathway inhibitors and novel antibody–drug conjugates. Furthermore, we delve into the role of immunotherapeutic agents such as interleukins as well as immune-promoting agents such as oncolytic viruses and cancer vaccines. Innovative combination therapies using these agents have led to a rapidly evolving treatment landscape and promising results for patients with recurrent ovarian cancer.

**Keywords:** ovarian cancer; resistance; poly (ADP-ribose) polymerase inhibition; antibody–drug conjugate; nanoparticle; checkpoint inhibition; vaccines; oncolytic viruses; monoclonal antibody; homologous recombination deficiency

## 1. Introduction

Ovarian cancer, which encompasses primary peritoneal and fallopian tube malignancies, is the deadliest gynecologic malignancy, resulting in an estimated 13,270 deaths in 2023 [1]. Ovarian cancer most often presents at an advanced stage due to a lack of effective screening modalities and the non-specific symptoms and signs of early stage disease. Alterations in key genes that are involved in DNA repair, known as homologous recombination repair (HRR) genes, play a critical role in the development of ovarian cancer. HRR genes, such as the breast cancer gene (*BRCA*) 1 and 2, are responsible for repairing DNA double-strand breaks and interstrand crosslinks. *BRCA* alterations are present in the tumors of approximately 25% of patients with newly diagnosed ovarian cancer [2]. Homologous recombination deficiency (HRD) refers to an inability to effectively repair DNA double-strand breaks using the HRR pathway. PARP inhibitors prevent cancer cells from repairing damaged DNA by blocking base excision repair, resulting in a buildup of DNA single-strand breaks that cannot be repaired by the deficient HR pathway, thus forcing the cancer cells to undergo apoptosis, with relative sparing of healthy cells [3]. Deleterious *BRCA* alterations and HRD status have been shown to predict improved progression-free survival (PFS) and overall survival (OS) in patients with ovarian cancer treated with PARP inhibitors [4]. Numerous assays to identify HRD are under investigation to aid treatment decision making [5]. As with the inevitable development of platinum resistance, we are increasingly presented with the challenge of how best to treat acquired PARP inhibitor resistance [6]. There is also an urgent unmet need to develop effective targeted therapeutic strategies for patients with HR-proficient ovarian cancer. A subset of

these patients harbor tumors with a *BRCA*ness phenotype, which have molecular traits similar to those of *BRCA*-mutated tumors [7] and are sensitive to DNA-damaging agents, including PARP inhibitors.

The role of immunotherapy in ovarian cancer continues to evolve. Immune checkpoint blockade modulates effector T-cell response by inhibiting the negative feedback mechanism and allowing T cells to successfully attack cancer cells. Various immunotherapeutic agents have shown promise in other solid tumors, including lung cancer and melanoma. Programmed cell death protein 1 and programmed death-ligand 1 (PD1/PD-L1) inhibitors allow for the persistent activity of cytotoxic T cells against cancer cells. Cytotoxic T-lymphocyte-associated protein 4 (CTLA-4) inhibitors are another class of immune modulators that promote the activation of cytotoxic T cells by preventing the binding of CTLA-4 to naïve T cells, causing senescence. In general, antibodies against PD1/PD-L1 are less toxic than anti-CTLA-4 agents. Unfortunately, these treatments have not shown efficacy as monotherapies in ovarian cancer [8], possibly due to intratumoral heterogeneity, which causes incongruent patterns of T-cell infiltration.

The future of immunotherapy in the treatment of recurrent ovarian cancer will likely rely on combination approaches; however, we have yet to identify the optimal immuno-oncologic approach. A randomized phase 2 trial assessed the combination of the CTLA-4 inhibitor ipilimumab with the PD-1 inhibitor nivolumab compared to nivolumab alone in recurrent ovarian cancer. Study findings showed that although the combination resulted in a slightly longer PFS (3.9 vs. 2 months, respectively; hazard ratio (HR) 0.53; 95% CI: 0.34–0.82) [9], with a small subset of patients deriving durable response, it was associated with increased toxicity. Similarly, a phase 2 study assessing the PD-1 inhibitor dostarlimab in combination with the PARP inhibitor niraparib in patients with recurrent, platinum-resistant ovarian cancer closed at interim analysis due to a low objective response rate of 7% [10].

Cancer vaccines designed to upregulate the immune response against ovarian cancer antigens are also under exploration [11]. Investigational protein or peptide-based vaccines target tumor-associated antigens such as cancer/testis antigens, differentiation antigens found predominantly in tumor cells but minimally expressed in normal tissue. These antigens are then recognized by effector T cells and generate a cytotoxic T-cell response, allowing for immune-mediated tumor killing [11]. Studies have also looked at dendritic cell-based vaccines armed with multiple tumor-associated antigens, which have shown promising clinical effects in patients with ovarian cancer [12].

Antibody–drug conjugates (ADCs) are novel therapies that employ monoclonal antibodies that target highly expressed tumor antigens to deliver cytotoxins directly into the cancer cells, inducing cell death with potentially decreased systemic toxicities. ADCs contain a linker that attaches the antibody to the potent cytotoxic agent. Folate receptor alpha (FRα) is an antigen found in approximately 60-90% of serous ovarian cancers [13,14]. FRα-targeting ADCs have already shown activity in ovarian cancer but have been associated with adverse events, such as keratitis, neuropathy, and interstitial lung disease [15]. Future efforts will focus on the optimization of ADCs as well as the exploration of novel combinations.

Platinum resistance has been associated with mutations in DNA repair pathways. These largely involve damage to sensory proteins that identify DNA breaks, including ataxia-telangiectasia mutated (ATM) and ATR (ATM- and Rad3-related) proteins. These proteins help maintain genomic integrity. It has been postulated that in platinum-resistant tumors, these DNA damage repair sensors are highly active, allowing for the reactivation of the cell cycle and tumor growth. Preclinical work has shown that targeting certain cell cycle checkpoints promotes sensitivity to PARP inhibitors and platinum agents, setting the stage for clinical trials. Another mechanism of chemotherapy resistance results from Axl receptor tyrosine kinase activation, leading to cell proliferation and survival by promoting epithelial–mesenchymal transition, which occurs when ligand GAS6 binds to Axl. Axl protein decoys have shown improved response rates in early phase clinical trials, offering new combination

therapies for platinum-resistant ovarian cancer. In this review, we describe some of the novel therapeutic approaches that are being investigated to treat this disease (Figure 1).

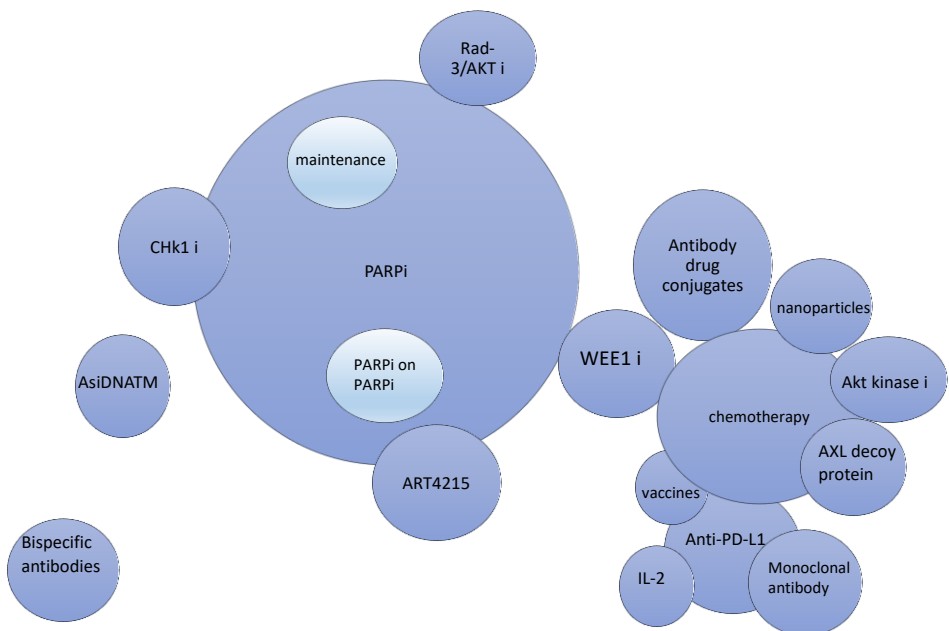

**Figure 1.** Schema of novel therapeutic agents (represented by individual spheres) and combination therapies that are currently FDA-approved or in the clinical trial phase. Combination therapies are represented by contact between the individual spheres. The size of the spheres indicates the number of clinical trials with the use of the respective agents.

## 2. The Current Role of Combination Chemotherapy in Ovarian Cancer

Despite advances in research, platinum-based chemotherapy remains the mainstay first-line treatment for ovarian cancer. Newly diagnosed high-grade serous ovarian cancer is considered chemosensitive, with response rates of up to 80% with the use of the carboplatin/paclitaxel doublet. Unfortunately, 80% of patients with ovarian cancer recur despite surgical cytoreduction and systemic platinum-based chemotherapy [16]. If recurrence occurs more than 6 months after the last dose of the platinum agent, the disease is deemed platinum-sensitive, and in the absence of a platinum allergy, further platinum-based therapy is recommended. Results from a phase 3 study in patients with recurrent, platinum-sensitive ovarian cancer showed improved PFS with gemcitabine plus carboplatin compared to single-agent carboplatin (8.6 vs. 5.8 months, respectively; $p$ = 0.003) [17]. In the CALYPSO trial, carboplatin and pegylated liposomal doxorubicin (PLD) improved PFS over carboplatin and paclitaxel in patients with recurrent, platinum-sensitive ovarian cancer (11.3 vs. 9.4 months, respectively; $p$ = 0.005) [18]. The addition of the anti-angiogenic bevacizumab to a platinum-based doublet compared to a platinum doublet alone has also shown improved survival outcomes in this setting [19]. In the OCEANS trial of carboplatin and gemcitabine +/− bevacizumab, the median PFS was 12.4 months for the triplet compared to 8.4 months for the doublet arm ($p$ < 0.0001), although the median OS was similar between the two groups (33.6 months and 32.9 months, respectively; HR 0.95; 95% CI: 0.77–1.18) [20].

In the platinum-resistant setting, defined as disease progression within 6 months of last treatment with a platinum agent, chemotherapy with single agents appears to result in similar response rates [21]. Findings from randomized phase 3 trials have shown PFS of 3–4 months and OS of approximately 12 months using either PLD, paclitaxel, topotecan, or gemcitabine [21]. The addition of bevacizumab led to a PFS increase of 3 months when administered in combination with paclitaxel, topotecan, or PLD in the AURELIA trial

(*p* = 0.001) [22]. Patients may also benefit from the addition of bevacizumab to gemcitabine in this setting [23].

### 3. PARP Inhibitors' Evolving Role in Ovarian Cancer Treatment and Maintenance

PARP inhibitor maintenance after response to first-line platinum-based therapy in patients with ovarian cancer with a somatic or germline *BRCA* mutation or HRD is now considered standard of care. The updated analysis of SOLO-1, a randomized phase 3 trial that evaluated the efficacy of olaparib as maintenance treatment in patients with newly diagnosed stage III or IV high-grade serous or endometrioid ovarian cancer with a *BRCA* mutation, found a significant increase in OS with olaparib compared to placebo (HR 0.55, 95% CI: 0.4–0.76; median OS was not reached vs. 75 months, respectively; *p* = 0.0004) [24]. In the PRIMA study, frontline maintenance therapy with niraparib was associated with a significant improvement in median PFS compared to placebo (21.9 months vs. 10.4 months, respectively; 95% CI: 0.31–0.59; *p* < 0.001) for patients with HRD, advanced high-grade serous or endometrioid ovarian cancer [25]. Although a PFS benefit was noted with niraparib compared to placebo in the overall population (13.8 vs. 8.2 months, respectively; 95% CI: 0.5–0.76; *p* < 0.001), the benefits were much less pronounced for those with HR-proficient tumors. Similarly, findings from the ATHENA study demonstrated improved median PFS with the PARP inhibitor rucaparib compared to placebo in patients with newly diagnosed advanced ovarian cancer after response to platinum chemotherapy (28.7 vs. 11.3 months, respectively; *p* = 0.0004) [26].

In the phase 3 ARIEL4 study, rucaparib was compared to chemotherapy in patients with recurrent *BRCA*-associated ovarian cancer. Patients in the rucaparib group had shorter median OS compared to those in the chemotherapy group (19.4 vs. 25.4 months, respectively; *p* = 0.0507) [27]. These results prompted the withdrawal of rucaparib as a treatment for recurrent ovarian cancer. Similarly, the results of the SOLO-3 trial demonstrated decreased median OS with olaparib compared to non-platinum chemotherapy in patients with recurrent germline *BRCA*-associated ovarian cancer (29.9 vs. 39.4, respectively; HR 1.33; 95% CI: 0.84–2.18). This also led to the withdrawal of olaparib as a treatment for patients with recurrent germline *BRCA*-mutated ovarian cancer. The NOVA study investigated the use of niraparib maintenance in the recurrent setting and found no significant difference in OS between patients who received niraparib and those who received placebo regardless of germline *BRCA* status, with a median OS of 40.9 months and 38.1 months, respectively (HR 0.85; 95% CI: 0.61–1.20) [28] These cumulative results led to its voluntary withdrawal as a treatment option for patients with recurrent HRD ovarian cancer.

PARP inhibitors are associated with an increased risk of therapy-related myeloid neoplasia, including myelodysplastic syndrome (MDS) and acute myeloid leukemia (AML). Patients are at increased risk for developing hematopoietic malignancies with prolonged PARP inhibitor treatment (more than 2 years), a history of extensive prior platinum exposure, and if they are *BRCA* mutation carriers. Patients treated with PARP inhibitors require ongoing monitoring for therapy-related myeloid neoplasia, even after cessation of the agent. Given this rare but lethal adverse event, there have been focused efforts to better identify patients most likely to derive benefit from PARP inhibitors and to identify those at the highest risk for developing AML/MDS.

### 4. DNA Replication and Repair Targets as Potential Treatment (Table 1)

Resistance to PARP inhibitors represents an ongoing challenge in the treatment of ovarian cancer [29]. One mechanism of resistance involves the restoration of genomic stability by protecting the replication fork, a crucial location where the DNA double helix is unwound and separated to allow for the synthesis of a new double helix. Novel targeted therapies have been developed to combat resistance to PARP inhibitors by reversing alternative pathways that protect the replication fork.

**Table 1.** Clinical trials using DNA replication and repair targets in high-grade serous ovarian cancer.

| Clinical Trial Number | Design | Target | Inclusion Criteria | Schema | Status |
|---|---|---|---|---|---|
| NCT03414047 | Phase 2 | CHK1 inhibitor (prexasertib) | HGSOC; Cohorts 1–3: platinum-resistant disease; Cohort 1: *BRCA* negative with 3 or more chemotherapy lines; Cohort 2: *BRCA* negative with fewer than 3 chemotherapy lines; Cohort 3: *BRCA* positive with prior PARP therapy; Cohort 4: platinum-refractory disease | Prexasertib administered IV compared with placebo | Completed |
| NCT05548296 | Phase 1b/2 | CHK1/2 inhibitor (ACR-368) | Platinum-resistant, advanced HGSOC/endometrioid ovarian cancer that has progressed on at least 1 prior regimen; also includes high-grade endometrial carcinoma and urothelial carcinoma | Participants with OncoSignature positive test administered ACR-368 as monotherapy; participants with a negative OncoSignature test administered a combination of ACR-368 and low-dose gemcitabine | Recruiting |
| NCT02264678 | Phase 1 | ATR and RAD-3 inhibitor (ceralasertib) | Solid tumors, including patients with prior PARP inhibitor use for certain modules, including *BRCA* mutations or HRD-positive status in certain modules | Ceralasertib + carboplatin versus ceralasertib + olaparib versus ceralasertib + durvalumab | Recruiting |
| NCT03682289 | Phase 2 | ATR and RAD-3 inhibitor (ceralasertib) | Solid tumors including renal cell carcinoma, urothelial carcinoma, pancreatic cancers, ovarian (excluding clear cell), and endometrial cancer | Ceralasertib alone versus ceralasertib + olaparib or ceralasertib + durvalumab | Recruiting |
| NCT04616534 | Phase 1 | ATR inhibitor (elimusertib) | Patients with pancreatic and ovarian tumors with measurable disease who progressed on at least one prior line of treatment | Elimusertib + gemcitabine | Recruiting |
| NCT04497116 | Phase 1/2a | ATR inhibitor (camonsertib) | Solid tumors resistant or refractory to standard treatment or patients with solid tumors who cannot tolerate standard therapy. Measurable disease as per RECIST v1.1 needed | Camonsertib alone versus camonsertib + talazoparib | Recruiting |
| NCT03462342 | Phase 2 | ATR inhibitor (AZD6738) | Recurrent ovarian cancer (platinum-sensitive or platinum-resistant) | AZD6738+ olaparib | Recruiting |
| NCT04991480 | Phase 1/2 | Polymerase theta inhibitor (ART 4215) | Advanced disease refractory to standard therapy; at least one radiologically evaluable lesion; estimated life expectancy >12 weeks | ART4215 as single therapy versus ART4215 with talazoparib, versus ART4215 with niraparib | Recruiting |
| NCT04826198 | Phase 1b/2 | DNA repair inhibitor (AsiDNATM) | Life expectancy of at least 3 months; availability of *BRCA* status; received at least 2 previous courses of platinum-containing therapy and has platinum-sensitive cancer; received niraparib in maintenance for at least 6 months | AsiDNATM in combination with niraparib versus AsiDNATM alone, niraparib alone, olaparib alone, and rucaparib alone | Recruiting |
| NCT04092270 | Phase 1 | DNA-PK inhibitor (peposertib) | Platinum-resistant or recurrent ovarian cancer | Peposertib with pegylated liposomal doxorubicin | Recruiting |

HGSOC: high-grade serous ovarian cancer; ATM: ataxia-telangiectasia mutated; ATR: ATM and RAD-3 related; CHK1: checkpoint kinase 1 CT: computed tomography; DNA: deoxyribonucleic acid; DNA-PK: deoxyribonucleic acid protein kinase; HRD: homologous recombination deficient; IV: intravenous.

### 4.1. Checkpoint Kinase 1 (CHK1)

CHK1 is part of the checkpoint signal transduction pathway and is activated in response to single-strand DNA damage [30]. It is also responsible for stabilizing the replication fork. If inhibited, cells that have sustained DNA damage will enter cell death. In a phase 1 study by Do et al., the addition of the CHK1/2 inhibitor prexasertib (LY2606368, ACR-368) potentiated sensitization to the PARP inhibitor olaparib in *BRCA*-deficient cancers [30]. In this study, 4 of 18 patients with *BRCA1/2* germline mutations and prior PARP inhibitor-treated high-grade ovarian cancer achieved a partial response. This set the stage for a phase 2 clinical trial assessing the efficacy of single-agent prexasertib in the treatment of recurrent ovarian cancer. One hundred and forty patients with platinum-resistant *BRCA*-wildtype or *BRCA*-mutated ovarian cancer demonstrated an overall response rate of 12% with prexasertib. The response rate was 7% in the additional cohort of 29 patients with platinum-refractory disease. No significant correlations were found between response to treatment and studied genomic alterations in cell cycle regulation or DNA damage response pathways (NCT03414047, [31]). Prexasertib was granted Fast Track designation by the Food and Drug Administration (FDA) in May 2023 given its promising early results. The agent is currently undergoing investigation with a novel companion test that measures the dependency of the tumor on activated CHK1/2 (NCT05548296). This phase 1b/2 trial is assessing its safety and efficacy as monotherapy and in combination with low-dose gemcitabine in platinum-resistant ovarian cancer.

### 4.2. ATR and RAD3-Related Inhibitors

PARP inhibitor resistance has inspired the use of other novel therapeutic agents that affect replication fork stability in patients with HR deficient tumors. In a phase 2 study by Mahdi et al., the use of ataxia telangectasia and the RAD3-related inhibitor ceralasertib in HRD and/or *BRCA*-mutated recurrent ovarian cancer in combination with olaparib led to an objective response rate of 8.3% and a clinical benefit rate of 62.5% [32]. Furthermore, of the seven patients with PARP inhibitor-resistant high-grade serous ovarian cancer, one achieved a partial response and five had stable disease [32].

### 4.3. Polymerase Theta Inhibitors

The polymerase theta inhibitor ART4215 is another novel agent under investigation for combating PARP inhibitor resistance, in this case by preventing reversion mutations. The polymerase theta inhibitor is under investigation with and without the PARP inhibitor talazoparib in an ongoing open-label phase 1/2a study in patients with advanced or metastatic solid tumors (including ovarian cancer) and germline or somatic defects in DNA repair (NCT04991480).

### 4.4. DNA Repair Inhibitors

The DNA repair inhibitor AsiDNATM is an oligonucleotide that acts as a decoy by mimicking double-strand DNA breaks, promoting apoptosis. It is now being assessed in a phase 1b/2 study in combination with PARP inhibition in patients with recurrent, platinum-sensitive ovarian cancer after at least 6 months of treatment with niraparib (NCT04826198).

Additionally, DNA-PK inhibitors, which also prevent the repair of DNA double-stranded breaks, block DNA-dependent protein kinase (DNA-PK), thus triggering apoptosis. Peposertib (M3814), an oral DNA-PK inhibitor, is being studied in ovarian cancer in conjunction with chemotherapy in a phase 1 trial (NCT04092270).

## 5. Mediators of Cell Cycle and Cell Signaling Pathways (Table 2)

Another mechanism of PARP inhibitor resistance is attributed to the restoration of the HR pathway, which is mediated by cell cycle and cyclin-dependent kinases. Targeting these checkpoints offers new opportunities for mitigating resistance.

**Table 2.** Clinical trials using cell cycle checkpoint inhibitors in high-grade serous ovarian cancer.

| Clinical Trial Number | Design | Target | Inclusion Criteria | Schema | Status |
|---|---|---|---|---|---|
| NCT04374630 | Phase 2 | AKT inhibitor (afuresertib) | HGSOC, endometrioid ovarian cancer, or ovarian clear cell carcinoma; no previous AKT or PI3K pathway or mTOR inhibitors; disease recurrence between 1 and 6 months after last dose of first-line platinum-based therapy or progression or relapse within 6 months of last dose of platinum-based second- to fifth-line therapies; 1–5 prior chemotherapies | Paclitaxel and afuresertib compared with paclitaxel | Active, not recruiting |
| NCT04729608 | Phase 3 | AXL decoy protein (batiraxcept) | Recurrent ovarian cancer (high-grade serous histology only); platinum-resistant disease; received at least 1 but no more than 4 prior therapy regimens | Batiraxcept in combination with paclitaxel versus placebo | Terminated |
| NCT05198804 | Phase 1/2 | WEE1 inhibitor (ZN-c3) | Recurrent high-grade epithelial ovarian cancer with histologic subtypes of serous, clear cell, or endometrial; recurrence within 6 months of platinum therapy | ZN-c3 administered with niraparib for 30 months | Recruiting |
| NCT04516447 | Phase 1b | WEE1 inhibitor (ZN-c3) | HGSOC, LVEF ≥ 50% | ZN-c3 in combination with liposomal doxorubicin, carboplatin, paclitaxel, or gemcitabine | Recruiting |
| NCT02993705 | Phase 3 | Alkylating agent (trabectedin) | Recurrent platinum-sensitive ovarian cancer with *BRCA1*/2 mutations or *BRCA*ness phenotype (patients who received and responded to at least 2 previous platinum-based treatments) | Trabectedin every 21 days versus physician's choice chemotherapy (carboplatin, gemcitabine, weekly paclitaxel, pegylated liposomal doxorubicin, or topotecan) | Completed |

HGSOC: high-grade serous ovarian cancer; LVEF: left ventricular ejection fraction

## 5.1. Pik3/AKT

PARP inhibitor resistance has also been associated with dysregulation of the PI3K/AKT checkpoint. In preclinical studies, AKT kinase inhibition was found to restore platinum sensitivity in patients with platinum-resistant ovarian cancer [33]. As a result, the pan-AKT inhibitor afuresertib is being evaluated in combination with paclitaxel in this setting (NCT04374630).

## 5.2. AXL

AXL is a tyrosine kinase receptor involved in the clearance of apoptotic cells and controlling the epithelial–mesenchymal transition. When AXL binds to its ligand Gas6, the cell is then driven into a proliferative state and protected against the immune response [34]. Batiraxcept (AVB-500), an AXL decoy protein, is designed to bind to its ligand growth arrest-specific 6 (Gas6) protein to prevent Gas6/Axl signaling. AVB-500 in combination with paclitaxel was associated with an objective response rate of 34.8%, including 2 complete responses among the 23 patients, with a median PFS of 3.1 months and OS of 10.3 months [35]. This combination was compared to paclitaxel monotherapy in a phase 3 clinical trial of 366 patients with platinum-resistant high-grade serous ovarian cancer (NCT04729608). Although no safety concerns were noted, the study was terminated due to a lack of significant improvement in PFS for the combination [36].

## 5.3. WEE1 Inhibitor

Cell-cycle checkpoint inhibition has also been evaluated as a potential therapy for ovarian cancer, especially given the high frequency of TP53 mutations in these cancers. The G2/M checkpoint is a crucial step in the cell cycle, assessing for DNA damage repair (NCT03579316) [37]. Although some initial promising results were seen with the WEE1 tyrosine kinase inhibitor adavosertib, the agent was associated with significant gastrointestinal toxicity and myelosuppression. The selective WEE1 inhibitor ZN-c3 is under investigation in patients with platinum-resistant ovarian cancer in combination with chemotherapy or PARP inhibition (NCT04516447 and NCT05198804, respectively).

## 6. Antibody–Drug Conjugates (ADCs) and Nanoparticle–Drug Conjugates as Potential Substitutes for Current Treatment (Table 3)

ADCs are novel biologic options that can target surface antigens found on ovarian cancer cells to deliver cytotoxic agents directly into cells.

**Table 3.** Clinical trials using antibody–drug conjugates/nanoparticle–drug conjugates in high-grade serous ovarian cancer.

| Clinical Trial Number | Design | Target | Inclusion Criteria | Schema | Status |
|---|---|---|---|---|---|
| NCT04296890 | Phase 3 | ADC against Folate Receptor α | Platinum-resistant HGSOC with high levels of folate receptor alpha | Single-agent mirvetuximab soravtansine on day 1 every 3 weeks until disease progression, unacceptable toxicity, withdrawal, or death | Completed |
| NCT03748186 | Phase 1 | ADC against Folate Receptor α | Patients with progressive or recurrent advanced epithelial ovarian carcinoma and endometrial cancer with a requirement to undergo folate receptor alpha testing | Luveltamab tazevibulin (STRO-002) administered once every 21 days in series of dose expansion | Recruiting |
| NCT04300556 | Phase 1/2 | ADC against Folate Receptor α | Platinum-resistant disease, triple-negative breast cancer, NSCLC, endometrial cancer, ovarian cancer | Administration of farletuzumab ecteribulin (MORAb-202) at 25 mg/m$^2$ and 33 mg/m$^2$ | Recruiting |

**Table 3.** *Cont.*

| Clinical Trial Number | Design | Target | Inclusion Criteria | Schema | Status |
|---|---|---|---|---|---|
| NCT04907968 | Phase 1 | ADC targeting sodium-dependent phosphate transport protein (NaPi2b) | Platinum-sensitive HGSOC | Combination of upifitamab, rilsodotin, and carboplatin administered every 28 days in patients with NaPi2b-positive HGSOC | Terminated |
| NCT05329545 | Phase 3 | ADC targeting sodium-dependent phosphate transport protein (NaPi2b) | Recurrent, platinum-sensitive HGSOC | Upifitamab rilsodotin administered as monotherapy versus placebo in patients with platinum-sensitive HGSOC with NaPi2b-positive disease | Terminated |
| NCT03587311 | Phase 2 | ADC with mesothelin antigen | Histologically or cytologically confirmed high-grade serous or high-grade endometrioid ovarian cancer with platinum-resistant or platinum-refractory disease and radiologic evidence of disease progression | Anetumab ravtansine administered with bevacizumab, and cycle repeated every 28 days in the absence of disease progression or toxicity, compared with paclitaxel and bevacizumab | Terminated |
| NCT04707248 | Phase 1 | ADC targeting Cadherin 6 | Patient with histological confirmation of advanced renal cell carcinoma or ovarian carcinoma with adequate cardiac function | R-Dxd administered once every 21 days in series of dose escalation to determine maximum tolerated dose | Recruiting |
| NCT04669002 | Phase 2a/b | Nanoparticle–drug conjugate | Advanced ovarian cancer, with platinum resistance; Cohort 1: more than 1 prior line of chemotherapy; Cohort 2: at least 1 prior line of chemotherapy followed by PARP inhibition for maintenance | EP0057 administered with olaparib (phase 2A) then EP0057 in combination with olaparib versus standard-of-care chemotherapy | Completed |

ADC: antibody–drug conjugate; HGSOC: high-grade serous ovarian cancer; NaPi2b: sodium-dependent phosphate transport protein.

### 6.1. Folate Receptor Alpha (FRα) ADCs

While there are several ADCs under investigation to treat gynecologic cancers, mirvetuximab soravtansine is the only ADC that is currently FDA-approved for ovarian cancer. Mirvetuximab is an ADC composed of an FRα-binding antibody, a cleavable linker, and a maytansinoid DM4 (a tubulin-targeting agent). The phase 2 SORAYA trial evaluated 105 patients with platinum-resistant, high-grade serous ovarian cancer with high FRα expression, defined as at least 75% of viable tumor cells expressing at least 2+ level membrane stain intensity on immunohistochemistry (IHC). Patients who had received 1 to 3 prior therapies including prior bevacizumab were randomized to mirvetuximab versus placebo. The study showed an overall response rate of 32.4%, including 5 complete and 29 partial responses, among the patients who received mirvetuximab soravtansine [14]. Findings from the confirmatory phase 3 MIRASOL trial, which compared the efficacy of mirvetuximab soravtansine to standard chemotherapy in platinum-resistant epithelial ovarian cancer, demonstrated an improvement in median PFS (5.6 vs. 4.0 months, respectively; $p = 0.0046$) [38]. The median OS in the mirvetuximab soravtansine group was 16.5 months (95% CI: 14.46–24.57), compared to 12.75 months (95% CI: 10.91–14.36) in the chemotherapy group (HR, 0.67; 95% CI: 0.5–0.89). The overall response rate was 42% for the FRα-ADC group versus 16% in the chemotherapy group ($p < 0.0001$). Given these positive findings, mirvetuximab soravtansine is now approved as a single agent for the treatment of platinum-resistant epithelial ovarian cancer with high expression of FRα.

In a recent study by Gilbert et al., mirvetuximab soravtansine was administered in combination with bevacizumab to patients with recurrent epithelial ovarian cancer expressing FRα (defined as at least 25% of tumor cells expressing at least 2+ level membrane stain intensity on IHC). The overall response rate was 44% (95% CI: 33–54), with 5 of 106 patients achieving a complete response. The median PFS was 8.2 months (95% CI: 6.8–10) [39].

Luveltamab tazevibulin (STRO-002) is another FRα-targeting ADC that has shown promise. It utilizes a stable cleavable linker and a 3-aminophenyl hemiasterlin warhead to induce cell death via site-specific conjugation. In a phase 1 trial that evaluated the efficacy of STRO-002 in platinum-resistant ovarian cancer regardless of FRα expression, objective responses were seen in 10 (32.2%) of 31 patients. Thirteen percent remained on treatment for over a year. An updated interim analysis from the ongoing phase 1 trial showed that 75% of patients achieved disease control (stable disease or partial response) (NCT03748186) [40]. This set the stage for FDA Fast Track designation for STRO-002.

Farletuzumab ecteribulin (MORAb-202) is a humanized monoclonal antibody directed against FRα. This ADC's payload is eribulin mesylate, a synthetic analog of halichondrin B that inhibits microtubules. In a small phase 1 study of 22 patients, 12 of whom had ovarian cancer, 1 patient achieved a complete response and 5 exhibited partial responses [13]. Overall, farletuzumab was well tolerated, with an acceptable toxicity profile and promising antitumor activity.

ADCs have unique toxicity profiles. The most common side effects are fatigue, nausea, and myelosuppression. Toxicity can vary according to the ADC and may depend on the payload and linker. Peripheral neuropathy, pneumonitis, and ocular toxicity (including keratopathy) can be encountered with mirvetuximab soravtansine. Mitigation strategies include the use of artificial tears and steroid-based eye drops as well as ophthalmologic assessment.

### 6.2. Sodium Phosphate Transport Protein ADC

NaPi2b is a sodium-dependent phosphate transport protein that can be overexpressed in ovarian cancer. The ADC upifitamab rilsodotin (also known as UpRi) targets this cancer-associated antigen. Unfortunately, the phase 3 clinical trial was terminated due to bleeding events. There were five grade 5 bleeding events among patients who received upifitamab rilsodotin (NCT05329545, NCT04907968) [41].

### 6.3. Mesothelin ADC

Mesothelin is a surface antigen that is overexpressed in 70% of ovarian cancers [42]. Anetumab ravtansine, a human antibody directed at mesothelin, is conjugated to a tubulin polymerization inhibitor, similar to the above ADCs. This compound was compared in combination with bevacizumab to weekly paclitaxel/bevacizumab in patients with platinum-resistant high-grade serous ovarian cancer (NCT03587311). Unfortunately, the ADC had limited efficacy, with only 1 complete and 4 partial responses compared to 16 partial responses in the weekly chemotherapy arm. The estimated median PFS was 5.3 months for anetumab ravtansine (95% CI: 3.7–7.4) and 9.6 months (95% CI: 7.4–17.4) for weekly paclitaxel (HR 1.7; 95% CI: 0.9–3.4) [43]. Infusion reactions and interstitial lung disease were observed.

### 6.4. Cadherin 6 ADC

Cadherin 6 (CDH6), a glycoprotein responsible for rapid internalization and cell-to-cell adhesion, is upregulated in ovarian and renal cancers [44]. A phase 1 study conducted by Hamilton et al. is evaluating the effect of DS-6000a, an ADC composed of humanized anti-CDH6 IgG1 monoclonal antibody linked to topoisomerase I, in patients with renal cell and ovarian cancers. An interim analysis reported a 46% overall response rate in patients with recurrent ovarian cancer (1 complete response and 22 partial responses) [45].

*6.5. Nanoparticle–Drug Conjugates*

Nanoparticle–drug conjugates are another novel treatment option for patients with recurrent, platinum-resistant ovarian cancer. The investigational drug EP0057 is composed of a cyclodextrin-based polymer backbone attached to camptothecin, a topoisomerase 1 inhibitor. Camptothecin stabilizes the topoisomerase-DNA complex during replication, leading to apoptosis of the cancer cells [46]. Data from a phase 1B/2 study demonstrated an overall response rate of 31.6% when combining the nanoparticle with chemotherapy, with 1 complete response among 19 patients. The median PFS for all patients was 5.4 months [46]. The study also demonstrated favorable pharmacokinetics for this novel drug, which could offer targeted therapy in select populations with recurrence.

**7. Bispecific Antibodies as an Immunotherapeutic Approach (Table 4)**

Bispecific antibodies can simultaneously bind a target tumor receptor on a cancer cell as well as surface markers such as CD3 on a T-cell to elicit tumor cell death. Ubamatamab (REGN4018) is a human bispecific antibody that binds mucin 16 (MUC16), a glycoprotein highly expressed in ovarian cancer cells, and the CD3 receptor on T cells, with the goal of inducing T-cell activation to kill ovarian cancer cells [47]. The overall response rate was 14% as monotherapy and 18.2% when administered in combination with the anti-PD-1 mono-clonal antibody cemiplimab in patients with recurrent ovarian cancer (NCT03564340) [48]. The median durations of response were 13.7 and 8.3 months, respectively.

**Table 4.** Clinical trial using bispecific antibodies in high-grade serous ovarian cancer.

| Clinical Trial Number | Design | Target | Inclusion Criteria | Schema | Status |
|---|---|---|---|---|---|
| NCT03564340 | Phase 1/2 | MUC16 × CD3 bispecific antibody (REGN4018) | Histologically or cytologically confirmed diagnosis of advanced, epithelial ovarian cancer, primary peritoneal or fallopian tube cancer with Ca-125 ≥ 2 × ULN; at least 1 line of platinum-containing therapy or must be platinum intolerant; documented relapse or progression of disease | REGN4018 administered in series of dose escalation, followed by administration alone and in combination with cemiplimab | Recruiting |
| NCT04590326 | Phase 1/2 | MUC16 × CD28 bispecific antibody (REGN5668) in combination with cemiplimab or REGN4018 | Histologically or cytologically confirmed diagnosis of advanced, epithelial ovarian cancer, primary peritoneal or fallopian tube cancer with Ca-125 ≥ 2 × ULN; at least 1 line of platinum-containing therapy | REGN5668 administered alone or in combinations with either cemiplimab or REGN4018 in series of dose escalation | Recruiting |

CD: cluster of differentiation; MUC-16: mucin 16; ULN: upper limit of normal.

REGN5668, a bispecific antibody against MUC16 and CD28, is being evaluated in combination with either cemiplimab or ubamatamab (NCT04590326). There was one confirmed partial response observed among the 22 patients treated in the REGN5668 and cemiplimab cohort during phase 1 dose escalation. The results from the dual bispecific antibody cohort are pending. These agents are associated with a risk for cytokine release syndrome, and similar to the ADCs, their toxicity profiles vary depending on the target antigen and drug construct.

**8. Immunotherapy in the Setting of Advanced, Recurrent Ovarian Cancer**

The role of immunotherapy in ovarian cancer has been difficult to delineate given the grim results from previous trials [8]. The reason for this has been largely attributed to the low immunogenic tumor microenvironment of ovarian cancer. However, recent combination regimens have given a new role for immunotherapy in the treatment of advanced ovarian cancer.

### 8.1. Immunotherapy with Chemotherapy

In a phase 2 trial assessing the efficacy and safety of pembrolizumab in combination with bevacizumab and oral cyclophosphamide in recurrent ovarian cancer (platinum-sensitive, -resistant, and -refractory), 3 patients (7.5%) achieved complete responses on this regimen, all of whom had platinum-resistant disease; 16 patients (40%) achieved partial responses (6 of whom had platinum-sensitive disease and 13 of whom had platinum-resistant disease); 19 (47.5%) had stable disease [49]. The objective response rate was 47.5%, the median PFS was 10 months, and 25% of patients experienced durable treatment responses [49]. Phase 3 trial data are pending.

### 8.2. Immunotherapy with Interleukin (IL)-2 Targets (Table 5)

Other novel immunotherapeutic combinations for the treatment of platinum-resistant ovarian cancer include pembrolizumab with nemvaleukin alfa, an engineered IL-2 variant. IL-2 has been shown to stimulate cytotoxic T-cell and natural killer cell growth and trafficking, potentially increasing sensitivity to immunotherapy. In the phase 1 ARTISTRY-1 trial, 5 of 15 patients with platinum-resistant ovarian cancer benefited from this combination treatment—1 patient experienced a complete response, 3 a partial response, and another patient had stable disease for over 1.5 years [50]. This combination is now under investigation in the phase 3 ARTISTRY-7 trial (NCT05092360).

**Table 5.** Clinical trials using cytokines in high-grade serous ovarian cancer.

| Clinical Trial Number | Design | Target | Inclusion Criteria | Schema | Status |
|---|---|---|---|---|---|
| NCT05092360 | Phase 3 | IL-2 (nemvaleukin alpha) and pembrolizumab | HGSOC, endometrioid of any grade, clear cell; platinum-resistant/refractory disease; at least 1 prior line of systemic anticancer therapy; at least one measurable lesion | Nemvaleukin and pembrolizumab compared with pembrolizumab alone, compared with nemvaleukin alone, compared with standard of care chemotherapy including either liposomal doxorubicin, paclitaxel, topotecan, or gemcitabine | Completed accrual |

IL: interleukin; HGSOC: high-grade serous ovarian cancer.

### 8.3. Immunotherapy with T-Cell Immunoreceptor with Ig and ITIM Domains (TIGIT) (Table 6)

Immunotherapy agents have also been paired with monoclonal antibodies, with the goal of promoting T-cell migration to the tumor microenvironment. The immune checkpoint molecule TIGIT has been shown to bind to the cell surface poliovirus receptor (PVR), leading to T-cell inactivation. Blocking TGIT promotes an amplified tumor response to prolong survival in patients with solid tumors [51]. Anti-TGIT antibodies have been paired with anti-PD1/PD-L1 agents, neoadjuvant chemotherapy, and other drugs to assess this response (NCT02794571, NCT05007106, NCT04570839, NCT04761198, NCT05026606, NCT 04254107).

Oregovomab is a murine monoclonal antibody that binds to CA-125. The hypothesis is that it would overcome tumor immunosuppression by attracting cytotoxic T cells to the cancer milieu. In a phase 2 study, treatment with oregovomab in combination with paclitaxel and carboplatin promoted a robust immune response and improved median PFS (41.8 vs. 12.3 months; HR 0.46; $p = 0.0027$) compared with paclitaxel and carboplatin alone in patients with newly diagnosed epithelial ovarian cancer. The median OS for the triplet was not reached versus 43.2 months (HR 0.35; $p = 0.043$) for the standard arm [52]. The FLORA-5 trial is currently recruiting patients for treatment with this regimen in the frontline setting (NCT04498117).

**Table 6.** Clinical trials using monoclonal antibodies in high-grade serous ovarian cancer.

| Clinical Trial Number | Design | Target | Inclusion Criteria | Schema | Status |
|---|---|---|---|---|---|
| NCT02794571 | Phase 1 | Monoclonal IgG1 antibody against TIGIT (tiragolumab) and anti-PD-L1 antibody (atezolizumab) | Locally advanced, recurrent, or metastatic incurable malignancy that has progressed after at least 1 available standard therapy | Tiragolumab alone or in combination with atezolizumab and tiragolumab; atezolizumab and tiragolumab with cisplatin and pemetrexed; atezolizumab and tiragolumab with carboplatin and pemetrexed; and carboplatin or cisplatin administered with etoposide after atezolizumab and tiragolumab; and atezolizumab and tiragolumab administered with capecitabine; atezolizumab and tiragolumab with bevacizumab; tiragolumab with pembrolizumab | Active, not recruiting |
| NCT05007106 | Phase 2 | Monoclonal IgG against TIGIT (vibostolimab) and anti-PD-L1 (pembrolizumab) | Ovarian, gastric, SCC, adenosquamous, adenocarcinoma of cervix, endometrial, head and neck, unresectable biliary adenocarcinoma, triple-negative breast cancer, adenocarcinoma, and SCC of esophagus, hepatocellular carcinoma, urothelial carcinoma | Pembrolizumab + vibostolimab versus pembrolizumab alone, versus pembrolizumab/ vibostolimab and lenvatinib versus pembrolizumab/vibostolimab with 5-flurouracil and cisplatin, versus pembrolizumab/vibostolimab with paclitaxel, versus pembrolizumab/vibostolimab with gemcitabine/cisplatin versus pembrolizumab/vibostolimab with carboplatin/paclitaxel/bevacizumab versus pembrolizumab/vibostolimab with capecitabine/oxaliplatin | Recruiting |
| NCT04570839 | Phase 1/2 | Monoclonal IgG against TIGIT (BMS-98207) and anti-PD-1 (nivolumab) and inhibitor of poliovirus receptor-related immunoglobulin domain (CPM701) | Locally advanced or metastatic solid malignancy; exhausted all available therapy; has not received prior therapy with an anti-PD-1 or anti-PD-L1, anti-CTLA-4, OX-40, CD137; platinum-refractory/resistant ovarian cancer | COM701 + BMS-986207 and nivolumab administered every 4 weeks | Active, not recruiting |
| NCT04761198 | Phase 1/2 | Monoclonal IgG against TIGIT (etiglimab) and anti-PD-1 (nivolumab) | Histological or cytological diagnosis of a relevant tumor type with available tumor tissue. Life expectancy >12 weeks and pre-specified wash-out of prior anti-PD-1/PD-L1 therapy | Etigilimab and nivolumab administered together every 2 weeks | Active, not recruiting |
| NCT05026606 | Phase 2 | Monoclonal IgG against TIGIT (etiglimab) and anti-PD-1 (nivolumab) | Recurrent clear cell ovarian cancer; platinum-resistant or -refractory disease; measurable disease on CT by RECIST | Etigilimab and nivolumab given every 28 days for up to 24 months in absence of disease | Recruiting |
| NCT04254107 | Phase 1 | Monoclonal IgG against TIGIT (SEA-TGT) and anti-PD-1 (sasanlimab) | Histologically or cytologically confirmed advanced or metastatic malignancy, including NSCLC, gastroesophageal carcinoma, cutaneous melanoma, bladder, cervical, ovarian, or triple-negative breast cancer, lymphomas | SEA-TGT given alone or in combination with sasanlimab | Active, not recruiting |
| NCT04498117 | Phase 3 | CA-125 antibody (oregovomab) | Newly diagnosed epithelial adenocarcinoma of ovarian, fallopian tube, or peritoneal origin; Stage III or IV; high-grade serous adenocarcinoma, high-grade endometrioid adenocarcinoma, undifferentiated carcinoma, clear cell adenocarcinoma, mixed epithelial carcinoma, or adenocarcinoma not otherwise specified; completed debulking surgery (must be optimal); preoperative CA-125 > 50; adequate bone marrow function | Oregovomab, paclitaxel, and carboplatin versus placebo, paclitaxel, and carboplatin | Active, not recruiting |

CD: cluster of differentiation; CTLA-4: cytotoxic T-lymphocyte-associated protein 4; IG: immunoglobulin; NSCLC: non-small cell lung cancer; PD-1: programmed cell death protein 1; PD-L1: programmed death-ligand 1; RECIST: response evaluation criteria in solid tumors; SCC: squamous cell carcinoma; TIGIT: T-cell immunoreceptor with Ig and ITIM domains.

## 9. Cancer Vaccines (Table 7)

The limited efficacy of immunotherapy in ovarian cancer may be partially attributable to low tumor immunogenicity. Vaccines have been created to boost the neoantigen response in an effort to overcome the hostile immune-sparse tumor microenvironment.

**Table 7.** Clinical trials using vaccines in high-grade serous ovarian cancer.

| Clinical Trial Number | Design | Target | Inclusion Criteria | Schema | Status |
|---|---|---|---|---|---|
| NCT04713514 | Phase 2 | Multi-neoepitope vaccine (OSE2101) | HLA-A2 phenotype, histologically or cytologically proven non-mucinous epithelial ovarian cancer; ECOG performance 0–1; first or second clinical or radiological recurrence of a platinum-sensitive ovarian cancer in complete response, partial response, or stable disease at end of platinum-based chemotherapy; previously treated with PARP inhibitor and not eligible for PARP inhibitor; prior therapy with bevacizumab or with contraindications to bevacizumab | Best supportive care versus OSE2101 versus OSE2101+ pembrolizumab as maintenance | Recruiting |
| NCT03029403 | Phase 2 | Vaccine targeting survivin (DPX-Survivac) | Histologically or cytologically confirmed advanced epithelial ovarian cancer; received platinum-based regimen following primary debulking or interval debulking with disease recurrence; radiologically documented disease progression | Combination of pembrolizumab, DPX-Survivac, and low-dose cyclophosphamide every 21 days | Active, not recruiting |
| NCT05104515 | Phase 1 | Vaccine targeting survivin (OVM-200) | Histologically confirmed metastatic or locally advanced inoperable NSCLC, ovarian cancer, or prostate cancer; at least 1 line of approved cancer therapy and either: exhausted current recognized treatment options or stable in planned treatment-free interval following completion of a set course of treatment; at least 1 measurable lesion | Dose escalation with 4 increasing doses of OVM-200. Phase 1b will then assess new dose in 3 expansion cohorts | Recruiting |

ECOG: Eastern Cooperative Oncology Group; HLA: human leukocyte antigen; PARP: poly (ADP-ribose) polymerase.

OSE2101, a vaccine with multiple epitopes targeting tumor-associated antigens CEA, p53, HER-2, MAGE-A2, and MAGE-A3, has been studied in various solid tumors and is now being evaluated in combination with pembrolizumab in platinum-sensitive recurrent ovarian cancer (NCT04713515). OVM-200 is a vaccine that targets survivin, an inhibitor of the apoptosis protein family. Survivin is overexpressed in ovarian cancer. OVM-200 has shown promising phase 1/2 results in breast, lung, colorectal, and ovarian cancer. The agent is now entering a phase 3 clinical trial (NCT05104515). A phase 2 study is investigating another survivin-targeting vaccine, DPX-Survivac, in conjunction with cyclophosphamide and pembrolizumab in recurrent ovarian cancer (NCT03029403). Both anti-survivin vaccines work by upregulating the cytotoxic immune response.

## 10. Oncolytic Viruses (Table 8)

Oncolytic viruses have also been shown to upregulate the immune response and promote powerful tumor-specific immunity. Oncolytic viruses directly attack cancer cells, causing lysis and release of tumor antigens, which promote a cytotoxic T-cell response. Olvi-Vec, which is made from the vaccinia virus, is administered as two consecutive intraperitoneal infusions to patients with platinum-resistant and -refractory ovarian cancer, and then followed by platinum-based chemotherapy and bevacizumab. In the non-randomized phase 2 study, the overall response rate was 54% (95% CI: 33–74%), with 8% achieving a complete response, 46% achieving a partial response, and 33% having stable disease. Tumor studies showed significant upregulation of cytotoxic T cells ($p = 0.008$) [53]. A phase 3 randomized

trial is underway (NCT05281471). Another oncolytic virus, TILT-123, is currently under investigation in combination with pembrolizumab for the treatment of platinum-resistant or -refractory ovarian cancer (NCT0527318).

**Table 8.** Clinical trials using oncolytic viruses in high-grade serous ovarian cancer.

| Clinical Trial Number | Design | Target | Inclusion Criteria | Schema | Status |
|---|---|---|---|---|---|
| NCT05281471 | Phase 1/2/3 | Vaccinia virus (Olvi-vec) and chemotherapy, bevacizumab | Histologically confirmed, non-resectable ovarian cancer, high-grade serous with metastasis; received a minimum of 3 prior lines | Olvi-Vec with carboplatin/paclitaxel and bevacizumab versus carbo/cisplatin and bevacizumab | Recruiting |
| NCT05271318 | Phase 1 | Oncolytic adenovirus (TILT-123) and anti-PD-1 (pembrolizumab) | Histologically confirmed ovarian cancer resistant to platinum or refractory to platinum; at least 1 tumor or carcinomatosis must be available for local virus injection (intratumoral/intraperitoneal) | Combination of TILT-23 and pembrolizumab, with dose escalation of TILT-123 | Recruiting |

PD-1: programmed cell death protein 1.

## 11. Discussion

Despite incremental improvements in ovarian cancer survival, most patients will experience a recurrence and subsequently develop chemotherapy resistance. Agents targeting DNA repair and cell cycle progression pathways represent new potential therapeutic options. Additionally, combination treatments are being investigated to overcome resistance and improve tumor killing by targeting various pathways simultaneously. Genomic testing is essential in the treatment of ovarian cancer. Additionally, tumor molecular alterations permit a more personalized approach in the selection of patients most likely to derive benefit from a particular therapy. Targeted therapies tailored to the patient's unique tumor biology are gaining ground, as can be noted with the expanding role of ADCs.

## 12. Conclusions and Future Directions

The treatment landscape of ovarian cancer is rapidly evolving, now encompassing investigations of various checkpoint inhibitors, ADCs, nanoparticles, and bispecific antibodies, as well as a better understanding of those most likely to benefit from PARP inhibitors. The goal of these new therapies is to combat acquired resistance by addressing various DNA stress-related pathways and unique cancer-associated targets. Novel diagnostic tools employing tumor genomics, cell-free DNA, IHC for target expression, and HRD testing are necessary to further stratify patients according to their tumor profile to allow for optimal targeted and individualized therapy.

**Key Points**

- The role of PARP inhibitors is rapidly evolving with new opportunities for combination treatment approaches;
- Targets of DNA repair pathways and checkpoint inhibitors are being investigated to overcome chemotherapy and PARP inhibitor-resistant ovarian cancer;
- ADCs allow for the targeting of novel cancer-associated antigens and more selective cytotoxicity.

Vaccines and bispecific antibodies can potentially augment the presence of cytotoxic T cells in the low immunogenic ovarian cancer tumor microenvironment.

**Author Contributions:** Conceptualization, data curation, writing—original draft preparation, review and editing—both authors. All authors have read and agreed to the published version of the manuscript.

**Funding:** Roisin E. O'Cearbhaill is supported in part by a Cancer Center Support Grant by the NIH/NCI (P30 CA008748).

**Conflicts of Interest:** Outside of this work Roisin E. O'Cearbhaill reports meeting/travel support from the Gynecologic Oncology Foundation, Curio, and Hitech Health; participation in the advisory boards of Tesaro/GlaxoSmithKline (GSK), Immunogen, Regeneron, Seattle Genetics, Fresenius Kabi, Bayer, and CarinaBiotech (non-compensated); non-compensated steering committee participation for Tesaro/GSK and AstraZeneca; and grant support paid to the institution from Alkermes, Bayer/Celgene/Juno, Lyell Therapeutics, Tesaro/GSK, Merck, the Ludwig Cancer Institute, Abbvie/StemCentrx, Regeneron, TCR2 Therapeutics, Marker Therapeutics, Syndax Pharmaceuticals, Genmab/Seagen Therapeutics, Sellas Therapeutics, Genentech, KitePharma, and the Gynecologic Oncology Foundation. Sara Moufarrij does not have potential conflicts of interest to disclose.

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
