# Peer review of "Novel Therapeutics in Ovarian Cancer: Expanding the Toolbox"

_curroncol, doi:10.3390/curroncol31010007_

Round 1

Reviewer 1 Report

Comments and Suggestions for Authors

Thank you for asking me to review this manuscript. The authors have listed many different treatment strategies and identified areas of interest for future study of ovarian cancer. This is a good summary and would be informative and of interest to a general reader. However, the actual text contains many errors in phrasing, misuse of words or poor sentence construction. I have included several examples below, but this is not a complete list.

Comments:

1)       “Abstract: Despite high response rates to initial therapy, most patients with ovarian cancer will ultimately develop treatment-refractory disease and will recur” – need to modify the opening to maintain logic sequence: patients have a recurrence first, then go on to get treatment resistance….or state that they develop treatment-refractory disease and progress or die. The same sentence structure is used in the discussion.

2)      “The role of immunotherapy in ovarian cancer is also expanding.” I think this statement is premature. To date, there is no definitive role of IO in “ovarian” cancer based upon phase 3 data (including the data presented for DUO-O), with no regulatory approvals. ADCs, although linked to an antibody, are not immunotherapy as the MOA is still primarily through direct DNA cytotoxicity.

3)      Line 259: the SORAYA trial was a phase 2 study.

4)      Given that the MIRASOL trial has led to a change in the SOC, more information on this trial and the results would be helpful to the reader. The manuscript should mention the RR, the OS impact and can consider mentioning the impact of prior Bev therapy, as the confirmatory trial did not require prior Bev usage. In addition, a comment about FR expression levels – how they are defined – could lower levels of expression be targeted? A comment on the ongoing studies of this agent in platinum-sensitive disease would make sense. Finally, what about a quick word on special toxicities?

5)      The MORAb-202 agent is mentioned, a phase 2 trial described…if this is a promising agent, are there ongoing trials? Please elaborate as the agent does not appear to be promising if no additional clinical testing is happening/planned.

6)      In general, the authors have several times described an agent as “promising” but either given data to demonstrate minimal efficacy or even inferior results to chemo (e.g. Mesothelin ADC) or have not given an indication about future clinical development of the agent. In some cases, it is stated that the agent is promising, but development was stopped due to toxicity (upifitamab rilsodotin). I think this has to be reconciled such that the wording does not conflict with the state of the drug in its development.

7)      “Napi2b, a sodium-dependent phosphate transport protein that can be overexpressed 282 in ovarian cancer, is another promising ADC for the treatment of ovarian cancer.” The Napi2b transport protein is incorrectly referred to as the ADC.

8)      Line 405: “Researchers have strived to counteract this resistance by targeting novel oncolytic pathways that are upregulated in the cancer microenvironment, as well as promote a cytotoxic immune response, which is commonly ablated in ovarian cancer.” The use of “oncolytic pathways” is not correct. “Oncolytic” typically refers to the selective targeting of tumour cells leading to cell death, but not to pathways. And “ablation” is a removal of something (by surgery or other means) and is not the correct term in this context.

9)      The conclusion refers to “BRCAness” despite not having explored this in the body of the manuscript.

Comments on the Quality of English Language

as above

Reviewer 2 Report

Comments and Suggestions for Authors

In the manuscript entitled “Novel therapeutics in ovarian cancer: Expanding the toolbox”, Moufarrij and O’Cearbhaill reviewed the novel treatment for ovarian cancer. The emerging therapy landscape will be of considerable importance in the field. However, I have listed below a series of questions and suggestions that could further improve the quality of the manuscript. These concerns must be addressed before the publication of this paper.

The Abstract reads more like introduction. It would be clearer if the authors directly summarized what will be reviewed. Please consider revising and better matching the review.

Line 35, PARP or PARPi are not for repairing interstrand crosslinks. Especially, the next sentence mentioned PARPi is for blocking base excision repair. Please consider refining.

In 4. DNA repair targets as potential treatment, why the authors mention Pik3/AKTand AXL pathway? Are they related to DNA repair? Instead, the authors should talk about ATM inhibitor and DNA-PK inhibitor and their related clinical trials.

For this section, it would be helpful if the authors could also write an introduction paragraph before moving into subtitles, just like sections 5 and 7.

The authors didn’t cite Table 1 and Figure 1 in the manuscript. It would be more organized if the authors could separate the Table1 into different tables and cite them within different subtitles.

Figure 1 must have legends to annotate why and how the model is depicted and what they mean. Or please explain and cite in respective section.

Reviewer 3 Report

Comments and Suggestions for Authors

Moufarrij and O'Cearbhaill have comprehensively reviewed the current and emerging therapeutics in ovarian cancer. The main text is well written and well-articulated. However, the authors need to elaborate the Table 1, where the actual status of the clinical trial such as recruiting or completed need to be indicated. The status of clinical trails could be reported in a separate column. Also, several ongoing trials just for ATR inhibitors such as NCT03682289,  NCT04616534,  NCT04497116,  NCT03462342  could be included and other ongoing trials for other targets need to be obtained from NCBI-ClinicalTrials.gov.

Comments on the Quality of English Language

The quality of English language is fine.

Author Response

Response: Thank you for the suggestion. We agree and have added the status of the clinical trials as a separate column in all the tables and have added additional ATR inhibitor trials to Table 1.

We have also elaborated more on the bispecific antibodies and added information on REGN4018 and REGN5668.

Round 2

Reviewer 1 Report

Comments and Suggestions for Authors

Thank you for the revisions provided. This is a comprehensive and current review that I believe will be of interest to many readers. 

Reviewer 2 Report

Comments and Suggestions for Authors

The authors have largely responded and addressed the major concerns. The manuscript quality was significantly improved. Here, I have no hesitation to suggest the publication of this paper.

Reviewer 3 Report

Comments and Suggestions for Authors

The authors have addressed the issues raised during first review. Therefore, I agree for the publication of this manuscript.